# Typology of Victimization against Women on Adolescent Girls in Three Contexts: Dating Offline, Dating Online, and Sexual Harassment Online

**DOI:** 10.3390/ijerph191811774

**Published:** 2022-09-18

**Authors:** María José Díaz-Aguado, Rosario Martínez-Arias, Laia Falcón

**Affiliations:** 1Unidad de Psicología Preventiva, Facultad de Psicología, Universidad Complutense de Madrid, Pozuelo de Alarcón, 28223 Madrid, Spain; 2Departamento de Ciencias de la Comunicación Aplicada, Facultad de Ciencias de la Información, Universidad Complutense de Madrid, 28040 Madrid, Spain

**Keywords:** adolescence, violence against women, victimization, sexual harassment online, gendered violence, justification of male dominance and violence, risky sexual online behaviors, self-esteem, feminine gender role stress

## Abstract

Defining the typologies of adolescent girls in relation to different types of victimization against women could be very useful for prevention. Almost all the typologies previously elaborated on this topic define the typologies from situations of dating victimization. This study used cluster analysis to establish for the first time a typology of adolescent girl victimization against women that included dating violence offline, dating violence online, and sexual harassment online outside a relationship by means of a comparative analysis of behavior between those who had suffered this violence and the population at large. The participants were 3.532 Spanish teenage girls aged 14–18 with experience of relationships with boys. Three discrete, identifiable types were obtained: the first group (63.8%), non-victim girls; the second group (29.4%), victims of sexual harassment online outside a relationship but with a low incidence of dating victimization; the third group (6.8%), victims in the three contexts. The logistic regression analysis showed that risky sexual behavior online was the main risk condition for inclusion in the second and third groups (compared to the non-victim group), followed by low self-esteem (for the second group) and age (for both groups). Other variables that also contributed to predicting membership victim groups were health complaints, feminine gender role stress, justification of male dominance and violence, visiting risky websites, and problematic internet use. These results show the importance of including the prevention of such problems in order to eradicate violence against women in adolescence who have grown up with digital technologies.

## 1. Introduction

As the United Nations recognizes [1], “achieving gender equality and empowering all women and girls is not only a fundamental human right but a necessary foundation for a peaceful, prosperous, and sustainable world”. One of the main obstacles that must be overcome to reach these goals is to eradicate violence against women and girls, “a widespread, persistent, and devastating problem that cuts across all generations, nationalities, and communities”.

### 1.1. Dating Victimization against Women and Girls from a Gendered Perspective

According to the World Health Organization [2], “worldwide, 27% of women aged 15–49 who have been in a relationship report having been subjected to some form of physical and/or sexual violence by their intimate partner”, a prevalence almost reached by adolescent girls. The European Union Agency for Fundamental Rights’ study [3] emphasized that around 43% of women aged 15–74 in Europe had suffered some form of violence, if the most frequent manifestations of this violence (such as psychological violence and control) were included. Studies of adolescents showed that adolescent dating victimization against women (DVAW) already may be present from the earliest relationships, and relates to other forms of victimization [4]; it can predict intimate partner victimization in adulthood [5] and has serious consequences for the victim’s development [6].

### 1.2. An Ecological Approach to Conditions of Risk and Protection

In order to prevent violence against women, it is important to consider the multiple contexts within which risk and protection conditions exist [7,8,9,10,11]. The analysis of such violence from an ecological perspective [12,13] considers characteristics derived from (1) individuals, (2) interpersonal relationships or microsystem, (3) the community or mesosystem, and (4) the society or macrosystem. According to the ecological approach to human development, the influence of playing a social role (such as the submissive role of a girl in a relationship with a boy) is evident at different contextual levels: the individual and microsystem levels where the interpersonal relationships take place, as well as at the community and macrosystem levels in which the role is rooted [12,13]. This reveals the relevance of representative research of the population, in order to study violence against adolescent girls in relation to the cultural context in which such problem occurs [14]. 

### 1.3. The Male Dominance Mentality Underlying Violence against Women

Among the main risk conditions for violence against women, in which the cultural context and the individual interact, the most important is the model of male dominance and feminine submission [6,8,15,16,17,18]. Studies conducted with adolescent girls have found a link between the justification of male dominance over women and adolescent DVAW, although this link is weaker than that found in male adolescent perpetration of violence against women [4].

### 1.4. Health Problems 

As the World Health Organization recognizes, violence against women and girls causes serious damage to their physical and mental health by producing (1) complex responses (neurological, neuroendocrine, and immune) to chronic stress, (2) risk behaviors to cope with this stress, and (3) problems derived from the abusive control they suffer, which hinders their access to health resources and other social contexts that would favor their development [2]. In relation to this, battered women syndrome, a subcategory of post-traumatic stress disorder, includes as its symptoms body image distortion and other somatic concerns [19,20].

### 1.5. Feminine Gender Role Stress (FGRS) and Self-Esteem

Women who are the victims of violence tend to have lower self-esteem than women and girls who have not suffered such violence [4,19]. This problem is also detected in adolescent girls exposed to gendered violence against their mother, which increases the risk of similar violence occurring in their own dating relationships [21].

According to the gender role strain theory, sexist stereotypes could lead to significant limitations to psychological development and wellbeing. Transgressions (actual or imagined) of gender roles may originate gender role stress and could lead to over-conformity to the stereotypes [22]. Most of the research on this problem has been conducted with men, confirming that masculine gender role stress predicts—both in adults [23] and in adolescents [24]—perpetration of violence against women to a greater extend than other components of sexism. Despite recognizing the need to analyze the relationship between gender role stress and the serious problems experienced by women [25,26], very little research has been carried out in this regard, although some investigations confirmed that FGRS is related to eating problems [27] and anxiety [28]. One of the aims of this research was to analyze the relationship between FGRS of adolescent girls and DVAW. 

### 1.6. Digital Technologies as a New Habitat 

Digital technologies have transformed teenagers’ lives both as an opportunity and as a risk. They can be used to facilitate relationships but also as a tool to exercise violence. In this sense, it has been found that experiencing offline dating abuse was by far the strongest correlate to experiencing online dating abuse [29].

The risks derived from digital technologies increase in the case of teenagers who use them in a more impulsive way [30]. Durkee et al. [31] found, in a study across 11 European countries, that almost all adolescents (90%) with problematic internet use (PIU) developed multiple risk behaviors. The relationship between PIU and adolescent DVAW is revealed in the fact that girls who reported having suffered this type of violence presented higher levels of PIU [4].

### 1.7. Sexual Harassment against Women Online

Among the most frequent problems regarding digital native teenagers stands out the reception of unwanted sexual solicitation or sexual harassment online, with 11% of women in the EU older than 15 reporting having been exposed to this problem. The figure for those aged 15–29 was twice that of the 40–49 age group, and three times more than those aged 50–59 [3]. Ferragut et al. [32], in research developed in Spain on the child sexual abuse recalled by adults aged 18–74, concluded that most forms of such abuse have decreased in the younger generation (18–24 years old) but that exposure to pornographic material and solicitations of personal photographs or videos with sexual content has increased. This is probably because such a new form of sexual abuse is suffered online, and teenagers use digital devices more frequently than previous generations. The relevance of this new problem has boosted research on adolescent sexual harassment online in the last decade, underlining that it is one of the most frequent abuses, and that girls are the victims in the majority of cases [4,33,34,35]. According to the Council of Europe’s definition of violence against women, such over-representation of girls among the victims of sexual harassment online leads its definition as a new form of violence against women [36]. In order to support the study on such violence from a gender-based approach, it must also be understood that the behaviors of sexual harassment online are perceived very differently by girls and boys; it has a far more negative emotional impact on girls, while its severity is downplayed by some boys [7]. Sexual harassment against women (SHAW) is clearly linked to risk behaviors that include visiting high-risk websites and meeting offline with contacts first made online [34,37,38].

### 1.8. Typologies of Girls in Relation to Victimization against Women

Defining the typologies of adolescent girls in relation to victimization against women could be very useful for prevention [4,39,40]. Almost all the typologies of adolescent girls on this topic define the typology according to the DVAW situations that they have suffered, which have yielded the following results:

Three types were detected in terms of seriousness and frequency of the DVAW situations suffered by the girls: a clear majority situation with few or no incidents of victimization, an intermediate situation, and an extreme situation, the latter suffered by a small minority of victims of multiple and frequent forms of DVAW [4,39,40,41]. The typologies in relation to the perpetration of DVAW that boys recognize having exercised also reflect three situations that differ according to the frequency and seriousness of the violence experienced [4,8,16,22,42]. These results show that these typologies can be very useful when detecting diverse DVAW’s risk levels, on the basis of which strategies to prevent this violence can be adapted.The types of situations detected in the typologies show that psychological victimization might exist where physical victimization is absent, but that serious physical victimization exists in conjunction with psychological victimization [4,39,40,41]. The typologies that evaluated situations of sexual abuse with four or more elements found that girls in the intermediate group remain in this category for both situations of psychological and sexual victimization [40,41]. Girls in the extreme group who have suffered multiple and frequent DVAW present higher levels of emotional stress, impulsiveness, and externalization behaviors than in the two other groups, as well as lower levels of self-esteem and self-control [4,39,40,41]; such differences can relate to the serious consequences or chronic stress produced by the violence against women [2,20]. Some research found that girls in the intermediate victimization group scored higher in these problems than those in the group that reported no victimization [4,39,40], while other studies found no significant differences [41]. Most of the differences in risk conditions relate to the levels of victimization experienced by the three groups, the highest being the group with multiple DVAW, followed by the intermediate group and then the non-victimization group, although, in some cases, the intermediate group did not differ significantly from one of the two other groups. The group with multiple and frequent victimization justifies dating violence more than the two other groups [4,41], while the intermediate group justifies it more than the group that has suffered no victimization [4]. These results are similar to those found in the DVAW typologies that boys acknowledge to have perpetrated [4,8], with the association between justification of gendered violence and the DVAW exercised by boys far greater than with the association of such mentality and DVAW suffered by girls [4,8].

Despite the groups of victims of DVAW experiencing more victimization in other contexts than the non-victim group, such as childhood sexual abuse [4,40] or sexual harassment online outside a relationship [4], we only found one typology of adolescent girls based on violence suffered in two different types of relation: childhood maltreatment and dating victimization [43]. This research with boys and girls in Denmark found four types: (1) revictimization group, with both childhood abuse and adolescent dating victimization, including sexual victimization (5.3%); (2) emotional and physical childhood abuse group (13.2%); (3) adolescent dating victimization group (19.4%); (4) low-victimization group (62.2%). Adolescent girls were more likely to be members of all three victimization groups than boys. The results of this study reflect that the inclusion of different types of victimization in elaborating the typologies can help identify the risk conditions with greater precision, as well as provide better specific protection for each group and, thus, structure prevention strategies in order to improve their efficacy [43].

We only found one typology developed from adolescent dating violence and online sexual abuse outside a relationship [8]. This typology is based on situations of abuse and violence that the boys admit to having committed, and which was formed of three types: the first group (69.8%), nonviolent boys; the second group (26%), more involved in online sexual harassment outside a relationship but with a low incidence of offline dating violence; the third group (4.2%), perpetrating abuse in the three contexts but less involved in online sexual harassment than the second group. Justification of male dominance and violence was the main risk condition for inclusion in the second and third groups, followed by low self-esteem (for the third group) and risky sexual behaviors online (for the second and third groups).

One of the conclusions of this research [8] is the need to investigate how the different types of male violence found in boys who perpetrate them combine with the perspective of the adolescent girls who suffer these types of violence as victims, and what risk and protection conditions (also for women) need to be considered for prevention, which is the aim of the research presented here. 

### 1.9. Victimization against Adolescent Girls in Spain

In order to comprehend the context of this research, it is helpful to consider the changes in equality between men and women that have occurred in Spain in last two decades. Such changes are especially relevant in the awareness and rejection of the specific violence perpetrated by men against women in a relationship. To counter this problem, a law was approved in 2004 [44], which became an international reference in this field. 

The study on adolescent violence against women in Spain [4] showed that the most common situations that girls reported experiencing were emotional abuse (insulting or ridiculing, 17.3%), general abusive control (deciding for me, down to the smallest detail, 17.1%), and control by mobile phone (14.9%). The percentage of boys who acknowledged perpetrating all types of DVAW was much less than that of girls who reported suffering them [8]. It is relevant to analyze the evolution of prevalence in adolescent DVAW in Spain in the last decade. Firstly, between 2010 and 2013, there was a worrying increase in this problem, probably linked to the emerging use of digital technologies to perpetrate DVAW, and the changes in teenagers’ relationships as a result of such technologies. In contrast, the period between 2013 and 2020 saw a marked decrease in adolescent girl DVAW, mainly in psychological abuse and control violence. Such change is linked to others detected among teenagers (less identification with the underlying mentality of gendered violence and greater identification with equality), inside the family context (fewer messages that support male domination of women, and more conversations between adults and adolescents on DVAW), and at school (more activities to build equality and prevent gendered violence) [4]. These changes detected in this study [4] relate to other important advances toward equality between men and women, and the struggle against gendered violence presented between 2013 and 2020 in the macro-system in Spain, such as greater scrutiny of this type of violence by the media, the 2017 cross-party parliamentary pact against gendered violence, or the mass demonstrations in support of equality that took place on 8 March 2018, which drew attention from the international press for their huge turnout [4].

### 1.10. Aims and Hypothesis

The main aims of this study were to define a typology of adolescent girls in relation to victimization against women that they reported having suffered in three contexts: dating victimization offline, dating victimization online, and sexual harassment online outside a relationship. Taking a general population sample enables us to understand how different contexts of victimization situations combine, and their relation to conditions that can be modified through preventive and palliative interventions: justification of male dominance and violence against women, feminine gender role stress, health complaints, problematic internet use, risk behaviors online, and self-esteem. 

The following hypotheses are proposed: 

There will be a substantial group of girls who show no kind of victimization (non-victim group), with the best risk and protective conditions, compared to the other two groups. One intermediate group will be identified, in comparison to the non-victim group, in terms of the victimization suffered, its health consequences, risk, and protective conditions. One small group will be identified, in comparison to the non-victim group, a minority of girls who have suffered victimization against women in three contexts, with the highest scores for the various risk conditions and for health complaints, as well as the lowest scores for self-esteem.

## 2. Materials and Methods

### 2.1. Design and Participants

The study was designed around a probabilistic sample survey with stratified two-stage cluster sampling. The primary sampling unit was the school, and the secondary unit was the classroom. Depending on the school size, one, two, or three full classrooms of students were randomly selected. The sample framework was the list of schools in Spain’s 18 autonomous regions provided by the regions’ educational authorities. The sampling design was stratified by region and type of secondary education center (compulsory, academic, and vocational) with sizes proportional to the population sizes. In the Spanish education system, secondary education is divided into compulsory (12–16 years old) and noncompulsory (17–18 years old), the latter divided between academic and vocational. To determine the effective sample size controlling the possible effects of intraschool resemblance, an intraclass correlation of 0.10 was considered. In practice, the effect of school on the main variables (DVAW offline 0.06; multiple DVAW online, 0.09; DVAW with messages online, 0.05; sexual harassment online outside a relationship, 0.04). Consequently, the design effect was not corrected in the statistical analysis. 

The initial sample consisted of 5395 Spanish female adolescents, of which those with no dating experience with boys (determined by an explicit question in the questionnaire) were not selected, as one of the objectives was to study the dating violence of boys against girls. The final sample included 3532 teenage girls aged 14 to 18 (M age = 15.90, SD = 1.22). The mean age of the first date was 13.33 (SD = 2.05).

The participants were enrolled at 222 secondary schools, 1966 (55.7%) in compulsory education, and 1566 (44.3%) in noncompulsory secondary education (1146, or 32.5%, in academic; 420, or 11.9%, in vocational). The mean number of participants per school was 16, ranging from eight to 90, with a median of 12. The number of participants attending state schools was 2425 (68.7%), and the participants from private schools numbered 1107 (31.3%). A total of 2875 students (91.6%) reported that they were native-born. Regarding the level of education of parents, 3175 students reported the level of education of their mothers and 3048 that of their fathers. In the case of the mothers, 56.5% had completed secondary education and 30.2% had a university qualification. Regarding the fathers, 58.0% had completed secondary education and 25% had a university qualification.

### 2.2. Procedure

The principals of the schools selected were informed of the study and their participation requested.

Consent procedures were put in place in accordance with recommended ethical guidelines. Parents of students under 18 had the opportunity to refuse consent for their child’s participation by returning a written form to the school office. Informed consent was requested from the 18 year old students. All students were instructed that the survey was voluntary, that they could withdraw at any time, and that their responses were anonymous.

Data collection at the school was carried out via internet. A teacher remained in the room as the survey was administered to answer questions and resolve potential computer problems. The average time required to complete the questionnaire was about 50 min. 

### 2.3. Measures 

All the measures used in this study were deployed and validated in previous research with Spanish adolescent girls [4,8], and the technical aspects and detailed psychometric properties of the measures are available for inspection [4].

#### 2.3.1. Adolescent DVAW Offline

The Adolescent DVAW offline questionnaire was composed of 10 indicators that referred to different forms of dating violence through control, emotional, physical, and sexual abuse. The indicators were (a) insults, (b) humiliation, (c) trying to isolate her from her friends, (d) trying to control all her behavior and decisions, (e) intimidating her, (f) physical aggression, (g) threats of aggression to force her to do things, (h) verbal intimidation or insults of a sexual nature, (i) pressure to perform sexual acts, and (j) accusing her of provoking the violence inflicted on her in any of the above situations. The response format was a four-point Likert scale: never, sometimes, frequently, many times [4] (p. 34). An exploratory factor analysis based on polychoric correlations was carried out with FACTOR 12.01.02 software [45]. An unweighted least-squares extraction produced one identifiable factor explaining 74% of the variance, with loadings greater than 0.70. The ordinal Cronbach alpha coefficient for the 10 items was 0.96, and the ordinal McDonald’s omega was 0.96. All the unidimensionality indicators showed suitable values. The summative score of the 10 items was used.

#### 2.3.2. Adolescent DVAW Online

This variable was evaluated using two scales.

*Multiple DVAW Online*. This was evaluated by five items with a four-point Likert-type response format: never, sometimes, frequently, and many times: (a) control via mobile phone, (b) use of passwords to exercise control, (c) use of passwords to impersonate, (d) sending messages to insult, threaten, offend, or frighten, and (e) spreading messages, insults, or images online. An exploratory factor analysis based on polychoric correlations by unweighted least-squares extraction was carried out with FACTOR software, with one factor explaining 74% of the variance, and with loadings greater than 0.56. The ordinal Cronbach alpha coefficient for the five items was 0.91, and the ordinal McDonald’s omega was 0.92. All the unidimensionality indicators showed suitable values. The summative score of the five items was used.

*DVAW with Messages Online*. This was evaluated by six items with a four-point Likert-type response format: never, sometimes, frequently, and many times. The items referred to types of messages sent by the partner via the web in order to (a) ridicule her, (b) insult her, (c) scare her, (d) threaten aggression to force her to do things, (e) disseminate images of her of a sexual nature without permission, and (f) pressure her to perform sexual acts. Exploratory factor analysis based on polychoric correlations with unweighted least-squares extraction carried out with FACTOR software produced one identifiable factor explaining 90.5% of the variance, with loadings greater than 0.80. The ordinal Cronbach alpha coefficient for these factors was 0.98, as was the ordinal McDonald’s omega. All the unidimensionality indicators showed suitable values. The summative score of the five items was used.

#### 2.3.3. Sexual Harassment against Women (SHAW) Online

The Online Victimization Scale for adolescents [46] was adapted to gauge online sexual harassment outside a relationship; it contained six items, with a four-point Likert-type response format: never, sometimes, frequently, and many times. The items were: (a) I have been asked for cybersex online; (b) I have been asked to continue talking about sex after I had asked him to stop; (c) they have spread rumors about my sexual behavior; (d) I have been asked for sexy photos online; (e) I have been shown sexual images online; (f) I have received unsolicited sexual content in emails or messages. Exploratory factor analysis based on polychoric correlations with unweighted least-squares extraction carried out with FACTOR software produced one identifiable factor explaining 68.6% of the variance, and with loadings greater than 0.52. All the unidimensionality indicators showed suitable values. Both coefficients of internal consistency, alpha and omega ordinals, showed the value of 0.91. The summative score of the six items was used. 

#### 2.3.4. Self-Esteem

We used the 10 items of the Self-esteem Scale of Rosenberg [47] widely used in the study of this construct, whose psychometric features have been studied in depth worldwide. Validation studies support the one-dimensional nature of this scale, which exhibited an internal consistency of 0.82 in the study sample. The extent of agreement with the statements was scored on a four-point Likert scale, from 1 for totally disagree to 4 for totally agree. 

#### 2.3.5. Justification of Male Dominance and Violence (JMDV)

A scale to measure justification of male dominance and violence against women in a relationship, consisting of seven Likert-type items with four points (1–4), was used. An unweighted least-squares extraction with FACTOR software showed one factor explaining 73.1% of the variance, with loadings greater than 0.71. All the unidimensionality indicators showed suitable values. Both coefficients of internal consistency, alpha and omega ordinals, showed the value of 0.94. The summative score of the seven items was used. The items were the following: “for the sake of her children, a women who puts up with violence from her husband or partner should not report him to the police”; “if a woman has been abused by her partner, she must have done something to provoke him”; “a proper father should make his family know that he is the boss”; “if a woman is battered by her partner and she does not leave him, it must surely be because she is not entirely unhappy in such a situation”; “for a relationship between a man and a woman to prosper, the woman should avoid contradicting her partner”; “the violence that takes place at home is a family matter and should be kept in the family”; “a man is justified in assaulting his wife or girlfriend when she decides to leave him”.

#### 2.3.6. Feminine Gender Role Stress (FGRS)

This variable was evaluated by 12 Likert-type items with five points selected from the 18 items of two factors from the Gillespie and Eisler Scale [22], with some modifications to adapt the wording to adolescents: the eight items of the physical unattractiveness factor and four items of the unemotional relationship factor. Exploratory factor analysis based on polychoric correlations with unweighted least-squares extraction carried out with FACTOR software produced two identifiable factors explaining 54.1% of the variance, with loadings greater than 0.40. The first factor, feminine gender role stress by physical unattractiveness (FGRS-PU), was composed of four items included in the same factor of the original scale: “being judged by others as overweight (too fat)”; “finding out that you gained 10 pounds (5 kg)”; “feeling less attractive than you once were”; “wearing a bathing suit in public”. The ordinal alpha coefficient reached the value of 0.79. The second factor, feminine gender role stress by unemotional relationship (FGRS-UR), included four items of this factor and four items of the physical unattractiveness factor of the original scale. The ordinal alpha coefficient value was 0.81. The items were the following: “not being able to meet family members’ emotional needs”; “your mate will not discuss your relationship problems with you”; “being considered promiscuous”; “having others believe that you are emotionally cold”; “being heavier than your mate”; “being unusually tall (taller than your boyfriend)”; “being unable to change your appearance to please someone”; “turning middle-aged (50 years old) and being single”. The summative scores were used for both factors. The correlation between factors was 0.718.

#### 2.3.7. Subjective Health Complaints (SHCs) 

This variable was evaluated by the Subjective Health Complaints Scale [48], with 11 Likert-type items with four points that referred to symptoms experienced by the individual with or without a defined diagnosis. Such symptoms constitute both everyday experiences and health problems, are common causes of disability, sickness, and absences from school [48], and arise as a result of exposure to gendered violence [49]. Exploratory factor analysis based on polychoric correlations with unweighted least-squares extraction carried out with FACTOR software produced two identifiable factors explaining 56.7% of the variance, with loadings greater than 0.40. The first factor, SHC-somatic composed of five items: “headache”, “abdominal pain”, “backache”, “dizziness”, and “neck ache”. The ordinal alpha coefficient value was 0.744. The second factor, SHC-psychological, consisted of six items: “feeling low”, “irritable”, “nervous”, “sleeping difficulties”, “exhausted”, and “afraid”. The ordinal alpha coefficient value was 0.856. The summative scores were used for both factors. The correlation between factors was 0.754.

#### 2.3.8. Problematic Internet Use (PIU) 

The Caplan Generalized Problematic Internet Use Scale 2 (GPIUS2) [50] was used to assess problematic internet use. The scale can be used in two ways, as a set of separate subscales or as a summative total score of the 15 items. The summative score was used. The ordinal alpha coefficient of the data from the study sample was 0.88.

#### 2.3.9. Risky Online Behavior

This scale was initially constructed according to indicators proposed by Ybarra et al. [38] on four types of risky online behavior: (1) “disclosure of personal information”; (2) “aggressive behavior”; (3) “interacting with someone you have met online”; (4) “risky sexual online behavior”. Exploratory factor analysis was carried out with FACTOR software [45]), using unweighted least-squares and Promin rotation as extraction methods. Three correlated factors were obtained that explained 56% of the variance: factor 1, disclosure of personal information (DPI), with six items: (a) “giving your first name and surnames to a stranger”, (b) “giving your home address”, (c) “accepting a stranger as an online friend”, (d) “giving the name of your school”, (e) “giving your age”, and (f) “giving your location”. The ordinal alpha coefficient for the set of six items was 0.845; factor 2, risky sexual online behavior (RSOB), with six items—three risky behaviors involving sexting, and three behaviors relating to interacting with someone you have met online: (a) “posting or sending a highly sexual photo of me”, (b) “posting or sending a sexual photo of my partner”, (c) “posting or sending a photo of me that my parents would not approve of”, (d) “talking about sex with someone I have met on the internet”, (e) “arranging to meet someone I have met on the internet”, and (f) “using webcam or mobile phone camera when I communicate with a stranger”. The ordinal alpha coefficient for the set of six items was 0.758; factor 3: visiting risky websites and aggressive behavior (VRWAB), with six items: (a) “browsing a website that my parents would not approve of”, (b) “browsing websites that show violence”, (c) “browsing websites that show sexual acts”, (d) “sending messages that insult or offend people”, (e) “phoning someone to upset them”, and (f) “responding to a message that has insulted or offended me”. The ordinal alpha coefficient for this set of six items was 0.784.

### 2.4. Data Analysis 

Data preparation was carried out using IBM SPSS v.28 software (Armonk, NY, USA). All analyses were made with SPSS v.28, except for the exploratory factor analysis of some scales that were conducted using FACTOR v.12.01.02 software [45]. Before computing the summative scores, the items’ missing values were imputed using IBM SPSS v.28 software with the expectation–maximization (E–M) algorithm. The procedure is iterative, and other variables are used to attribute a value (expectation) followed by checks to see whether that is the most likely value (maximization). If not, it re-imputes a more likely value. This continues until the most likely value is reached [51].

The main procedure used for the data analysis was two-step clustering that classifies individuals into groups on the basis of their patterns of responses to sets of observed variables. It is an exploratory multivariate data analysis that identifies similar groups of objects within datasets with both continuous and categorical variables [52]. This study used the likelihood distance measure as the similarity criterion and the Bayesian information criterion (BIC) to determine the optimal number of clusters. Two-step cluster analysis also calculates the silhouette coefficient, which measures the quality of the clustering solution. Validity is established on the basis of a combination of two different measures: cohesion, which is the closeness between the members of the same cluster, and separation, which is the closeness between the members and the centroids of the different clusters. The procedure is performed in two distinct phases. In the first step, a sequential clustering procedure is used to create pre-clusters by scanning the data case by case and assigning them to a previously formed cluster or a new one, according to the distance criterion. When all cases have been assigned to a pre-cluster, all objects in the same pre-cluster are treated as a single entity to reduce the size of the matrix that contains distances between all possible pairs of cases [52]. The second step consists of applying a hierarchical clustering algorithm to the pre-clusters and defining the best number of clusters based on a BIC. 

Clusters were formed from four standardized quantitative variables of adolescent victimization against women: DVAW offline, multiple DVAW online, DVAW with messages online, and SHAW online outside a relationship.

Due to the lack of compliance with the assumptions of the parametric tests, the relationship between cluster membership and quantitative variables was examined using the Kruskal–Wallis (K–W) nonparametric test with all pairwise comparisons; significance values were adjusted by the Bonferroni correction for multiple tests. Statistical significance for the comparisons was set at 0.01 due to the large sample size. 

Rosenthal’s *r* statistic, with values between 0 and 1 [53], was used to compute the effect sizes of the pairwise comparisons. 

Lastly, the variables that showed statistically significant differences among groups were entered as predictors in a multinomial logistic regression analysis.

## 3. Results

Table 1 shows the correlations between variables used in the formation of clusters (triangular lower matrix) and the descriptive statistics. All relationships for variables included in the cluster analysis were positive and significant (*p* < 0.001).

The variables were previously standardized. The solution selected following the Schwarz-BIC was the three-cluster solution, which provided a good solution (close to 0.75, with 1.00 being the maximum value) in cohesion and separation. All variables were important in the cluster formation. 

The numbers of subjects in the clusters were 2254 (63.8%), 1040 (29.4%), and 238 (6.8%) for clusters 1, 2, and 3, respectively. Table 2 shows the main results for the four variables in the clusters (mean, standard deviation, range, and median), as well as the statistical significance of the differences.

The K–W test with two degrees of freedom showed statistically significant differences for each of the four variables used in cluster formation. Results for K–W and each of the comparisons were as follows: DVAW offline (K–W (2) = 1091.20, *p* < 0.001); multiple DVAW online (K–W (2) = 1394.41, *p* < 0.001; DVAW with messages online (K–W (2) = 1649.02, *p* < 0.001); SHAW outside a relationship (K–W (2) = 1858.55, *p* < 0.001. Pairwise statistically significant differences adjusted by the Bonferroni correction are presented in Table 2, as well as the effect sizes. 

According to the results in Table 2, members of the first group had means and medians that were very low on all variables; on the basis of the response pattern, this group was termed *non-victim* because almost none of the members had received abusive behavior from their male partners or online sexual harassment from men outside a relationship. Two groups of adolescents reported having received abusive behaviors to some extent. Members of the second group had received dating violence online and offline, but to a much lesser extent than in group 3. Group 2 stands out from the other two in SHAW outside a relationship. We named this group *victim of*
*sexual harassment online.* Group 3 of female adolescents scored high in dating victimization online and offline, as well as in sexual harassment online; on the basis of this response pattern, it was defined as *victim in the three contexts.*

Possible differences between clusters in terms of age and type of education were explored. Age revealed significant differences among groups with a very small effect size: F (2,3529) = 15.97, *p* < 0.001, *η*^2^ = 0.009. The Bonferroni post hoc contrast showed that subjects in groups 2 and 3 were older than those in group 1 (Ms = 15.94, 16.16, 16.23; *p* < 0.001). The chi-squared contrast showed a significant association between cluster and type of education, χ^2^ (4) = 54.13, *p* = 0.001, Cramer’s V = 0.09. Membership of clusters 2 and 3 was more frequent among vocational education students. 

The covariates with the cluster membership were those mentioned in the hypotheses: self-esteem, justification of male dominance and violence (JMDV), feminine gender role stress (as PU and UR), health complaints (somatic and psychological), PIU, and the three factors of online risk behaviors (DPI, RSOB, and VRWAB).

Table 3 shows the descriptive statistics for covariates, correlations, and collinearity. The collinearity statistics were computed and were within the established limits.

Table 4 presents the main results for the covariate variables in the clusters (mean, standard deviation, and median), the statistical significance of the pairwise comparisons, and effect sizes. 

The K–W test with two degrees of freedom showed statistically significant differences between clusters in all covariates: self-esteem (K–W = 144.45, *p* < 0.001), FGRS-PU (K–W = 149.74, *p* < 0.001), FGRS-UR (K–W = 116.87, *p* < 0.001), SHC-somatic (K–W = 125.76, *p* < 0.001), SHC-psychological (K–W = 216.33, *p* < 0.001), JMDV (K–W = 25.93, *p* < 0.001), PIU (K–W = 246.27, *p* < 0.001), DPI (K–W = 144.73, *p* < 0.001), RSOB (K–W = 661.17, *p* < 0.001), and VRWAB (K–W = 528.90, *p* < 0.001). 

The comparisons of medians tests were also carried out, with results similar to those of the K–W tests; hence, they are not presented here.

In most of the variables (self-esteem, FGRS-PU, SCH-somatic, PIU, DPI, RSOB, and VRWAB), cluster 1 was lower than clusters 2 and 3, with no differences between the latter two. The differences were statistically significant among the three groups in role stress by unemotional relationship (FGRE-UR) and SHC-psychological, showing the same pattern: cluster 1 < cluster 2 < cluster 3.

The results from the multinomial logistic regression are presented in Table 5. The reference for comparison was the *non-victim group* (cluster 1). Variables that were statistically significant for the prediction (*p* < 0.05) were age, self-esteem, FGRS-PU, SCH-somatic, JMDV, PIU, RSOB, and VRWAB. The final model was statistically significant, χ^2^ (22) = 4797.16, *p* < 0.001, indicating that at least one of the predictors in the model was not equal to zero. The Nagelkerke pseudo *R*^2^ value was 0.286. 

The odds ratios indicated the predicted change in the odds of membership of a particular class compared to the nonvictim group (cluster 1) for a one-standard-deviation increase in the covariate, with all other variables in the model remaining constant. 

First, predictors of membership of the *victim of sexual harassment online* cluster, compared to the *non-victim* cluster, were considered. An increase of one standard deviation in RSOB increased the odds of membership of this cluster by 26%, while, in the case of VRWAB, it increased the odds by 11%. Both FGRS-PU and SCH-somatic increased the odds of membership by 4%, while age increased it by 10%. Self-esteem decreased the odds of membership of this cluster by 17%. 

Next, predictors of membership of cluster 3, *victim in the three contexts*, compared to the *non-violent* cluster, were considered. Increases of one standard deviation in RSOB increased the odds of membership of cluster 3 by 28%, both VRWAB and JMDV increased the odds by 8%, and age increased the odds by 17%. SCH-somatic increased the odds of membership of cluster 3 by 7%, whereas this increase was 3% in the case of FUR, and 2% in the case of PIU. 

## 4. Discussion

Three discrete, identifiable types were obtained from the two-step cluster analysis based on victimization against women that the girls reported having suffered in three contexts: (1) DVAW offline; (2) DVAW online-1 and DVAW online-2; (3) SHAW online outside a relationship. As expected, the biggest group (first group, 63.8%) consisted of non-victim adolescent girls (hypothesis one), and the third group, a small minority of 6.8%, who identified with victimization in the three contexts, was formed of adolescent girls who frequently suffered victimization against women within a relationship in both offline and online scenarios, as well as sexual harassment online (hypothesis three). The analysis also found the existence of an intermediate group (second group, 29.4%), with a low incidence of DVAW offline and online, and as much sexual harassment online outside a relationship as the third group. This partially confirms hypothesis two. On the other hand, these results show that DVAW offline and DVAW online are closely related, as other studies also found [29].

In line with hypothesis three, the group of girls who had suffered multiple victimization in three contexts showed higher scores for all risk conditions and health problems (physical and psychological), as well as lower self-esteem than the non-victim group. These results can relate to those of previous investigations [4,20,41,42,44], with the serious consequences that the chronic stress generated by DVAW has for the physical and psychological health of the victims, and the greater probability of them developing risk behaviors to try to cope with this stress [2], especially when they have suffered violence in multiple contexts and relationships [29], together with their greater tendency to justify the violence they suffer [4,42], which makes it difficult for them to free themselves from this situation and increases the risk of revictimization [4,20]. The multinomial logistic regression analysis showed that the main predictors of belonging to this third group (compared to the non-victim group) were risky sexual online behavior and age (probably due to the link between age and more time spent in relationships and use of digital devices). Other problems that make it possible to predict belonging to this third group are justification of male dominance and violence, visiting risky websites, physical health complaints, feminine gender role stress by unemotional relationship, and problematic internet use.

The intermediate group, victims of sexual harassment online, showed lower self-esteem, higher scores for health problems (physical and psychological), and all risk conditions (with the exception of justification of male dominance and violence) than the group of non-victims. Fewer differences were observed between this intermediate group and the victimization group in the three contexts; in only three of the 10 problems evaluated was group 3 worse than group 2, as expected. The regression analysis showed that the main predictors of belonging to this intermediate group (compared to the group of non-victims) were risky sexual online behaviors and lower self-esteem, as well as, to a lesser extent, visiting risky websites, age, feminine gender role stress by physical unattractiveness, and physical health complaints. 

The fact that the main predictor of belonging to the two groups of victims was risky sexual online behavior is highly relevant for prevention. Although previous research detected the relationship between these behaviors and DVAW in adolescent girls [4,34,38,39], this research discovers it for the first time in a typology based on victimization against women in three contexts. It should be noted that this risky sexual online behavior factor is formed by three items involving interaction with someone they have met online and three risky sexting behaviors. As concluded by previous researchers [34], adolescents may have learned to normalize these behaviors by online interaction with strangers who proffer sexual solicitations as part of their persuasion strategy, justifying and portraying online sexual behaviors as natural.

As expected, the two groups of victims had lower self-esteem and more FGRS when they did not conform to sexist stereotypes (of physical attractiveness and affection in relationships) than the non-victim group. This could lead them to exaggerate traditional female submission to male domination and violence, the antithesis of the empowerment of women. 

The comparison of the results obtained in this research to those of the typology based on the violence against women that boys admit to having exercised [8] shows that, in both cases, three groups were detected, although the percentage of boys in groups with violence was somewhat lower than that of girls with victimization. Risky sexual online behaviors stand out as an important risk condition for violence against women, from what the boys acknowledge having exercised and the girls acknowledge having suffered. The main differences occur in the justification of male dominance and violence against women, with a much higher predictive value for violence of boys than for the victimization of girls, as well as with respect to age, for which the opposite is true. This last result should relate to the nature of such victimization, whose biggest risk conditions are linked to being a woman and the length of time exposed to situations that activate the model of male dominance over women [1], as occurs in the three contexts studied here.

Along with the strengths of this study (a typology of victimization against women from the perspective of girls who have suffered it in three contexts, with a large representative sample and the use of a non-victim group as reference group), this research also had some limitations that are important to consider. As the data are based on responses from self-reporting, they should be supplemented by other qualitative procedures. In this sense, future research could use focus groups and attempt to draw out the normative beliefs that underlie the normalization of risky sexual online behaviors. Secondly, a longitudinal study is needed to investigate the evolution of the problems detected here. Thirdly, it would be useful to investigate the role of attitudes and beliefs on sexual harassment online outside a relationship among the boys who perpetrate it and the girls who suffer it as victims.

## 5. Conclusions

The identification of an intermediate group, with sexual harassment online and its risk conditions, is one of the main contributions of this research. The high percentage of girls included in this group, as well as the health problems that characterize them, highlights the seriousness and extent of the problem of sexual harassment online, to which very little attention has been paid so far in terms of prevention of violence against women. To counter this problem, it is important to help adolescents understand that sexual harassment online is a new form of violence against women that can seriously harm its victims and denigrate those who perpetrate it. This goal could be especially difficult with boys who seem to use and accept such behaviors as a normal way to relate to women.

Another contribution of this research is its verification for the first time that feminine gender role stress (due to physical unattractiveness and unemotional relationships) increases the risk of victimization against women. Hence, the need to help girls overcome this emotional component of sexism, making them aware that the stereotype of the woman as an attractive object and the need to be liked and accepted by everyone is a sexist stereotype that is impossible to fully achieve, representing a serious obstacle to their development and wellbeing, and increasing the risk of submission and victimization. 

In relation to this is the fact that two of the characteristics that distinguish the two groups of victims when compared to the non-victim group, their lower self-esteem and the greater stress they feel by not conforming to sexist stereotypes, are the antithesis of the empowerment of women that the United Nations includes as one of the objectives for sustainable development in the 2030 agenda [1]. These results highlight the relevance that this objective continues to have in preventing violence against women in cultural contexts, even though great progress has been made toward equality. 

The fact that the main predictor of belonging to the two groups of victims is risky sexual online behavior reflects the need to prevent such behavior through digital literacy, raising awareness in adolescents as receivers and creators of online messages, and helping them to understand that people they have met online may not be what they appear to be, and that such risky sexual online behavior (such as sending sexual photos of themselves) can hand a potential harasser a weapon with which to victimize them. The relationship between victimization and visiting risky websites (such as pornographic pages) highlights the need for adequate affective–sexual education for equality, which reduces the motivation to seek information from sources that increase the risk of objectification of women and of sexual violence.

The fact that these results occur within a macrosystem such as Spain, where many important advances in the fight against violence against women have been made, goes to show that such gains have coincided with the emergence of new forms of violence against women by means of digital devices, which also needs to be countered by working on the main risk conditions.

## Figures and Tables

**Table 1 ijerph-19-11774-t001:** Descriptive statistics and correlations between variables used for the formation of clusters and descriptive statistics.

	Mean (SD)	Median	1	2	3	4
1 DVAW offline	11.76 (3.87)	10.00	-			
2 DVAW online-1	5.63 (1.71)	5.00	0.78 ***	-		
3 DVAW online-2	0.97 (3.25)	0.00	0.70 ***	0.75 ***	-	
4 SHAW online	4.70 (4.87)	3.00	0.29 ***	0.25 ***	0.25 ***	-

DVAW: dating victimization against women; online-1: multiple victimization online; online-2: with messages online; SHAW: sexual harassment against women online. *** *p* < 0.001.

**Table 2 ijerph-19-11774-t002:** Descriptive statistics of cluster variables and significant of the differences among clusters.

	Cluster 1: Non-Victim		Cluster 2: Victim of Sexual Harassment Online		Cluster 3: Victim in the Three Contexts		Pairwise, ES (*r*)
	Mean (SD) (Range)	Median	Mean (SD) (Range)	Median	Mean (SD) (Range)	Median	
DVAW offline	10.48 (1.03) (10–19)	10.00	12.00 (2.44) (10–23)	11.00	22.80 (6.98) (10–40)	22.00	(C1 < C2) ***, ES = 0.34 (C1 < C3) ***, ES = 0.61 (C2 < C3) ***, ES = 0.52
DVAW online-1	5.10 (0.35) (5–8)	5.00	5.69 (1.06) (5–11)	5.00	10.40 (3.44) (5–20)	10.00	(C1 < C2) ***, ES = 0.32 (C1 < C3) ***, ES = 0.70 (C2 < C3) ***, ES = 0.66
DVAW online-2	1.03 (0.13) (0–6)	0.00	1.03 (2.49) (0–10)	0.00	9.81 (6.55) (0–24)	10.00	(C1 = C2) (C1 < C3) ***, ES = 0.80 (C2 < C3) ***, ES = 0.81
SHAW online	1.90 (2.16) (0–7)	1.00	9.75 (4.03) (0–18)	10.00	9.10 (5.66) (0–18)	9.00	(C1 < C2) ***, ES = 0.72 (C1 < C3) ***, ES = 0.39 (C2 = C3)

DVAW: dating victimization against women; online-1: multiple victimization online; online-2: with messages online; SHAW: sexual harassment against women online. ES (*r*): Rosenthal’s *r* effect size; *** *p* < 0.001.

**Table 3 ijerph-19-11774-t003:** Descriptive statistics of covariates and correlations.

	Mean, (SD) (Range)	1	2	3	4	5	6	7	8	9	10
1 Self-es	28.59 (6.70) (10–40)	-									
2 FGRS-PU	6.31 (4.44) (0–16)	−0.037	-								
3 FGRS-UR	10.23 (6.18) (0–32)	−0.281 **	0.602 **	-							
4 SCH-1	7.48 (3.56) (0–15)	−0.262 **	0.224 **	0.191 **	-						
5 SCH-2	10.18 (4.45) (0–18)	−0.475 **	0.383 **	0.341 **	0.592 **	-					
6 JMDV	7.70 (1.79) (7–28)	−0.030	0.034	0.124 **	−0.010	0.001	-				
7 PIU	18.33 (11.48) (0–60)	−0.279 **	0.310 **	0.348 **	0.180 **	0.336 **	0.154 **	-			
8 DPI	9.28 (4.96) (0–18)	−0.032	0.138 **	0.146 **	0.163 **	0.186 **	0.029	0.267 **	-		
9 RSOB	2.11 (2.95) (0–18)	−0.205 **	0.191 **	0.169 **	0.148 **	0.234 **	0.158 **	0.364 **	0.316 **	-	
10 VRWAB	4.03 (3.63) (0–18)	−0.198 **	0.223 **	0.203 **	0.159 **	0.276 **	0.144 **	0.422 **	0.371 **	0.603 **	-

Self-es: self-esteem; FGRS-PU feminine gender role stress—physical unattractiveness, FGRS-UR: feminine gender role stress—unemotional relationships; SHC-1: subjective health complaints—somatic; SHC-2: subjective health complaints—psychological; JMDV: justification of male dominance and violence; PIU: problematic internet use; DPI: disclosure personal information; RSOB: risky sexual online behavior; VRWAB: visiting risky website and aggressive behavior; VIF: variance inflation factor; Tol: tolerance. ** *p* < 0.001.

**Table 4 ijerph-19-11774-t004:** Descriptive statistics of covariates in the three clusters, pairwise comparisons, and effect sizes.

Covariates	>Cluster 1: Nonvictim		Cluster 2: Victim of Sexual Online Harassment		Cluster 3: Victim in the Three Contexts		Pairwise Comparisons, ES (*r*)
	Mean (SD)	Median	Mean (SD)	Median	Mean (SD)	Median	
Self-es	29.65 (6.29)	30.00	26.84 (6.94)	27.00	26.20 (7.19)	26.00	(C1 > C2) ***, ES = 0.19; (C1 > C3) ***, ES = 0.15; (C2 = C3), ES = 0.01.
FGRS-PU	5.59 (4.17)	5.00	7.50 (4.55)	7.00	7.96 (4.87)	8.00	(C1 < C2) ***, ES = 0.12; (C1 < C3) ***, ES = 0.22; (C2 = C3), ES = 0.03.
FGRS-UR	9.38 (5.96)	9.00	11.48 (6.16)	11.00	12.82 (6.67)	12.00	(C1 < C2) ***, ES = 0.16; (C1 < C3) ***, ES = 0.17; (C2 < C3) *, ES = 0.07.
SHC-1-somatic	6.98 (3.49)	7.00	8.23 (3.48)	8.00	8.86 (3.57)	9.00	(C1 < C2) ***, ES = 0.14; (C1 < C3) ***, ES = 0.19; (C2 = C3), ES = 0.06.
SCH-2-psychol.	9.36 (4.33)	9.00	11.48 (4.25)	12.00	12.24 (4.34)	13.00	(C1 < C2) ***, ES = 0.22; (C1 < C3) ***, ES = 0.25; (C2 < C3) *, ES = 0.07.
JMDV	7.62 (1.63)	7.00	7.74 (1.68)	7.00	8.33 (3.01)	7.00	(C1 = C2), ES = 0.04; (C1 < C3) ***, ES = 0.09; (C2 < C3) **, ES = 0.09.
PIU	16.05 (10.60)	15.00	21.90 (11.34)	21.00	24.34 (13.70)	24.00	(C1 < C2) ***, ES = 0.16; (C1 < C3) ***, ES = 0.49; (C2 = C3), ES = 0.04.
DPI	8.52 (4.94)	8.00	10.63 (4.69)	11.00	10.56 (4.82)	10.50	(C1 < C2) ***, ES = 0.20; (C1 < C3) **, ES = 0.12; (C2 = C3), ES = 0.01.
RSOB	1.16 (1.95)	1.00	3.73 (3.52)	3.00	4.02 (3.95)	3.50	(C1 < C2) ***, ES = 0.41; (C1 < C3) ***, ES = 0.28; (C2 = C3), ES = 0.01.
VRWAB	2.95 (3.06)	2.00	5.94 (3.98)	5.00	5.96 (4.17)	5.00	(C1 < C2) ***, ES = 0.38; (C1 < C3) ***, ES = 0.23; (C2 = C3), ES = 0.01.

Self-es: self-esteem; FGRS-PU feminine gender role stress—physical unattractiveness, FGRS-UR: feminine gender role stress—unemotional relationships; SHC-1: subjective health complaints—somatic; SHC-2: subjective health complaints—psychological; JMDV: justification of male dominance and violence; PIU: problematic internet use; DPI: disclosure personal information; RSOB: risky sexual online behavior; VRWAB: visiting risky website and aggressive behavior. Kruskal–Wallis tests showed statistically significant differences, *p* < 0.001. ES (*r*): Rosenthal test of effect size; *** *p* < 0.001, ** *p* < 0.01, * *p* < 0.05.

**Table 5 ijerph-19-11774-t005:** Multinomial logistic regression with cluster membership as dependent variable.

	B (SE)	95% IC for Odds Ratio				95% IC for Odds Ratio		
		Lower	Odds Ratio	Upper	B (SE)	Lower	Odds Ratio	Upper
Cluster 2. Victim of Sexual Online Harassment					Cluster 3. Victim in the Three Contexts			
Intercept	−3.67 *** (0.69)				−7.52 *** (1.15)			
Age	0.09 * (0.04)	1.02	1.10	1.18	0.16 * (0.06)	1.04	1.17	1.32
Self-es	−0.19 * (0.07)	0.72	0.83	0.96	−0.16 (0.12)	0.67	0.85	1.08
FGRS-PU	0.04 ** (0.01)	1.01	1.04	1.07	0.03 (0.02)	0.99	1.03	1.07
FGRS-UR	0.01 (0.01)	0.99	1.01	1.02	0.03 * (0.01)	1.00	1.03	1.06
SCH-1	0.04 ** (0.02)	1.01	1.04	1.07	0.07 ** (0.02)	1.02	1.07	1.13
SCH-2	0.02 (0.01)	0.99	1.02	1.04	0.03 (0.02)	0.98	1.03	1.08
JMDV	−0.01 (0.03)	0.94	0.99	1.05	0.08 * (0.03)	1.01	1.08	1.15
PIU	0.02 (0.01)	1.00	1.01	1.02	0.02 * (0.01)	1.00	1.02	1.03
DPI	−0.01 (0.02)	0.96	1.01	1.03	−0.01 (0.02)	0.96	1.00	1.03
RSOB	0.23 ***(0.02)	1.21	1.26	1.31	0.25 *** (0.03)	1.21	1.28	1.36
VRWAB	0.11 *** (0.01)	1.08	1.11	1.15	0.07 ** (0.02)	1.03	1.08	1.13

Cluster 1 is the reference class. Self-es: self-esteem; FGRS-PU: feminine gender role stress—physical unattractiveness, FGRS-UR: feminine gender role stress—unemotional relationships; SHC-1: subjective health complaints—somatic; SHC-2: subjective health complaints—psychological; JMDV: justification of male dominance and violence; PIU: problematic internet use; DPI: disclosure personal information; RSOB: risky sexual online behavior; VRWAB: visiting risky website and aggressive behavior. *** *p* < 0.001, ** *p* < 0.01, * *p* < 0.05.

## Data Availability

The data presented in this study are available on reasonable request from the second author (rosmarti@ucm.es). The data are not publicly available due to ethical and privacy reasons.

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
