# Peer review of "Typology of Victimization against Women on Adolescent Girls in Three Contexts: Dating Offline, Dating Online, and Sexual Harassment Online"

_ijerph, 2022, doi:10.3390/ijerph191811774_

Round 1

Reviewer 1 Report

Dear Authors,

This is a welcome and interesting paper, well argued and thought provoking. The article presents a well-written, interesting, important and original contribution to the study of typology of violence against adolescent girls.

The research methodology chosen is explained clearly and the measures used in the study are described and validated. This reviewer also liked the ways in which the limitations of the research were clearly identified and the discussion of the implications of their research.

Although I think that your article is not quite ready for immediate publication, I think that it could be if the following revisions were made to it. The revisions will require a bit of additional time and effort, but I think this would be worthwhile in terms of the final quality of the completed article.

I say the following comments in a spirit of trying to help you. I’d very much like to see published this paper in this journal.

Congratulations !!!

INTRODUCTION

The subsections of the introduction explain interesting and necessary topics and are well referenced. However, they are too short.

It would be necessary to go more deeply into each subsection and to better show the relationship between them.

For expert authors the information may be sufficient, but for non-expert proffesional it may be more complicated to understand.

For example, sub-section 1.1. is too short for the topic it deals with. Also 1.2., the ecological model could be explained a bit more.

OBJECTIVES AND HYPOTHESES

It would be interesting to change the format of the objectives and hypotheses to make them clearer to read.

One option could be to list each objective and then show the hypotheses (also numbered, e.g. Hypo. 1, Hypo. 2).

PROCEDURE

could indicate something about the ethics committee?.

TABLE 4

I think that in self-es, in pairwise comparison, the value of ES is missing.

DISCUSSION

The beginning is too direct. A short introduction would be appropriate.

If changes are made to the objectives and hypotheses, it would be interesting to include the numbering in the discussion.

Author Response

Dear reviewer, 

Thank you very much for your work.

Kind regards,

Reviewer 2 Report

I found the paper to be quite solid and informative. The topic is relevant, meaningful, and important.

The paper needs a conclusion. Some of the information the discussion could be moved to the conclusion.  The conclusion should contain the "so what." The findings are important, so make sure the conclusion is equally important and useful for public policy. 

The use of "violence against women suffered by adolescent girls" is confusing throughout.  At first, I though the paper was related to sexual violence against girls perpetrated by women. You use Violence Against Women as a category of violence, with a subcategory of adolescents. I would change the title slightly and/or explain this out in the abstract and the body of the paper. Similarly, "Justification of Male Dominance" - needs explaining, as it reads as though the dominance is justified, which is not the intent. Justification by whom? Operationalize this if it is a standard reference in the field.

Line 188 - "one of the conclusions..." -start a new paragraph, as this is an important point and should not get lost.

Line38, change goal to goals

Author Response

Dear reviewer, 

thank you very much for your suggestions.

Kind regards,

Reviewer 3 Report

This manuscript addresses a very relevant and subjectViolence against Women Suffered by Adolescent Girls in Three Contexts. This is a very important and timely subject and is a potentially significant contribution to our field. In general, the manuscript follow the requirements of the journal and is very well written, organized and structured. However, there are some aspects that could be improved.

- The title is too long and would benefit from further systematization

- also the introduction is too long and should be systematized and focused

- it would be interesting and in view of the objectives of the study for the authors to present the research hypotheses

- Although the results are interesting and globally well discussed, I believe that the practical implications of the study deserve further development and discussion.

Author Response

(The authors gave the same response as above.)
